# Association between patient activation, self-management behaviours and clinical outcomes in adults with diabetes or related metabolic disorders: a systematic review and meta-analysis protocol

Julia Mueller [1], Amy L Ahern [1], Stephen J Sharp [1], Rebecca Richards [1], Jack M Birch [1], Alan Davies [2], Simon J Griffin[1,3]

¹MRC Epidemiology Unit, University of Cambridge, Cambridge, UK
²Division of Informatics, Imaging & Data Sciences, The University of Manchester, Manchester, UK
³Department of Public Health and Primary Care, University of Cambridge, Cambridge, UK

**Correspondence to**
Dr Julia Mueller;
julia.mueller@mrc-epid.cam.ac.uk

## ABSTRACT

**Introduction** Diabetes and related metabolic disorders such as obesity and cardiovascular diseases (CVD) are a growing global issue. Equipping individuals with the necessary 'knowledge, skills and confidence to self-manage their health' (ie, patient activation (PAct)) may lead to improvements in health outcomes. It is unclear whether existing evidence allows us to assume a causal relationship. We aim to synthesise and critically appraise evidence on the relationship between PAct and self-management behaviours and clinical outcomes of people living with diabetes and related metabolic disorders.

**Methods and analysis** The protocol is based on guidance on Preferred Reporting Items for Systematic Review and Meta-analysis Protocols. We will search Medline, Embase, CENTRAL, PsycInfo, Web of Science and CINAHL using search terms related to PAct, diabetes, pre-diabetes, obesity and CVD. Any quantitative study design is eligible provided studies assess the association between PAct and clinical outcomes and/or self-management behaviours of diabetes and related metabolic disorders. Outcomes include behavioural (eg, diet) and clinical (eg, blood pressure) outcomes. Two reviewers will independently screen titles/abstracts and full texts and assess risk of bias using the revised Cochrane risk-of-bias tool for randomised trials or the Risk of Bias Assessment Tool for Nonrandomised Studies (RoBANS).
One reviewer will extract data, with independent checking by a second reviewer. We will critically assess the level of evidence available for assuming a causal association between PAct and outcomes. Data permitting, we will use the Hunter-Schmidt random-effects method to meta-analyse correlations across studies.

**Ethics and dissemination** Ethical approval is not required. The review will be disseminated in the form of a peer-reviewed journal article, at conferences and other presentations. The findings of the review will be of interest to clinical commissioning groups, policymakers and intervention deliverers/developers.

**PROSPERO registration number** CRD42021230727.

---

### Strengths and limitations of this study

► This review assesses whether patient activation is a proxy measure for wider health outcomes and includes a broad range of clinical and behavioural outcomes.
► It uses a comprehensive search strategy with a broad range of relevant databases, including databases that allow insight into grey literature (eg, conference abstracts, theses).
► We will conduct a thorough critical appraisal of the evidence, based on a systematic procedure adapted from previous reviews, to assess whether evidence supports causal assumptions.
► We expect high heterogeneity across studies, which may make meta-analysis infeasible or difficult to interpret.

## BACKGROUND

Excess body weight is a major risk factor for chronic health problems such as diabetes mellitus and cardiovascular disease (CVD).[1 2] Diabetes and related metabolic disorders (eg, obesity and CVD) are linked to poor patient outcomes such as reduced quality of life[3] as well as increased direct and indirect economic costs, mainly due to medication, hospitalisations, disability and loss of productivity.[4–10] Equipping individuals with the necessary knowledge, skills and confidence to achieve sustained changes in their behaviour and self-manage their health and healthcare may lead to improvements in health-related outcomes and reduced hospitalisation and costs.[11–15]

The construct encompassing patients' knowledge, confidence and skills for self-management has been termed 'patient activation' (PAct).[16] Consumer-driven healthcare

**Table 1** Search terms for the systematic review

| Concept | Free text | MeSH |
|---|---|---|
| Patient activation | "patient* activation*" measure* ADJ5 "patient activation" PAM?22* PAM?13* PAM??13* PAM??22* "Patient Assessment of Chronic Illness Care*" PACIC* | |
| Diabetes | Diabet* T2DM T1DM (non insulin* depend* or non insulin depend* or non insulin?depend* or non insulin?depend) IDDM or NIDDM or MODY T1D or T2D | exp Diabetes Mellitus, Type 2/or exp Diabetes Mellitus/ or exp Diabetes Mellitus, Type 1 exp diabetes insipidus |
| Prediabetes | Pre?diabet* Borderline ADJ3 diabet* Impair* ADJ3 glucose "Non-diabetic hyperglyc?emi*" Glucose ADJ3 intoleran* | exp Prediabetic State/ or exp Glucose Intolerance/ |
| Obesity/Overweight | Obes* Overweight "over weight" Body ADJ3 weight "body weight" Adiposit* Weight adj3 (gain* or loss* or chang* or control* or maintain* or reduc* or manag*) Bmi or body mass ind* | exp Obesity/ OR exp Overweight/ OR exp Body Weight/ OR exp Adiposity/ or exp body mass index/ |
| Heart disease | Heart* OR cardiovascular OR coronary OR cardio* OR cardiac* | exp Heart Diseases/ OR exp Cardiovascular Diseases/ exp Coronary Disease/ OR exp heart failure/ |

approaches and many chronic illness care models assume that more 'activated' patients (ie, patients with the relevant knowledge, confidence and skills to self-manage their own health and healthcare) will play a more active role in managing their health and have better health outcomes.[16] Conversely, less 'activated' patients are expected to be less likely to see out help, adhere to medical advice and manage their own health. A recent systematic review on PAct in adults with chronic conditions identified two measures of PAct, the Patient Activation Measure (PAM) and Patient Assessment of Chronic Illness Care (PACIC), which includes a subdomain on PAct.[17] The PAM is the most commonly used instrument to assess PAct. It is a self-report measure with either 22 or 13 items (short form).[16 18] PAM scores range from 0 to 100 with higher scores, indicating higher activation. PAM scores are categorised by four stages of activation: stage 1 (≤47.0) and stage 2 (47.1–55.1) are categorised as low activation levels, and stage 3 (55.2–67.0) and stage 4 (≥67.1) are categorised as high activation levels. The PAM is widely used in healthcare delivery and evaluation.[19 20] For example, within the UK National Health Service (NHS), the PAM is used for population segmentation and risk stratification in order to target and tailor interventions.[19] General practitioner practices have used the PAM to tailor their diabetes review process such that participants with lower activation levels receive longer appointments than those with high activation levels.[20] PAM scores are also used to allocate different interventions to individuals with different activation levels. As such, it is important to understand how the PAM (and other PAct measures) are associated with clinical outcomes and self-management behaviours.

### PAct and self-management behaviours relevant to diabetes and related metabolic disorders

There is some evidence to indicate that PAct is associated with self-reported self-management behaviours relevant to diabetes and related metabolic disorders, such as eating a healthy diet, being physically active, adhering to medication and smoking cessation.[16 18 21–26] For some outcomes, such as self-reported physical activity, the relationship with PAct appears consistent.[16 18 21 22 24 25] For other outcomes, the relationship is less clear. For example, some studies have found no significant association between PAct and smoking,[21–23] and in Hibbard & Tusler's study, correlations with diet-related variables (eg, self-reported fruit and vegetable consumption) seemed to vary depending on the population and the specific behaviour measured.[22] Although several studies have assessed associations between PAct and self-management behaviours, this evidence has, to our knowledge, not been synthesised in a systematic review.

### PAct and clinical outcomes of diabetes and related metabolic disorders

Self-reported behavioural measures are prone to error (which may be correlated with error in the measure of

**Table 2** Risk of bias tools to be used in the review, depending on study design

| Study design | Risk of bias tool |
|---|---|
| Randomised controlled trial | RoB 2: a revised Cochrane risk-of-bias tool for randomised trials[49] |
| Observational studies | Risk of Bias Assessment Tool for Nonrandomised Studies (RoBANS)[50] |

*RCTs that have been analysed as a cohort study (ie, reporting on the association between PAct and outcomes, regardless of study group allocation) will be assessed using the RoBANS tool. If the data we extract depend on study group allocation, we will use the RoB 2 tool.
PAct, patient activation; RCT, randomised controlled trial.

PAct) and bias. Furthermore, it is not clear how associations between PAct and health behaviours translate into clinical outcomes. As the PAM is used in the evaluation of healthcare systems and interventions,[19] it is important to understand not only if this measure (and any other PAct measures) predict self-management behaviours (such as adhering to a healthy diet) but also how PAct measures relate to clinical outcomes.

Several studies have found significant associations between PAct and clinical outcomes such as haemoglobin $A_{1C}$ (HbA$_{1C}$), blood glucose, triglycerides, cholesterol and blood pressure.[23 26–30] However, the evidence base is heterogeneous and complex, with some studies finding no significant associations,[26 28] significant associations opposite to those hypothesised[26] or inconsistent patterns across PAct levels (ie, unclear dose–response relationships).[27] The relationship between PAct and objective clinical outcomes is, therefore, unclear and warrants further investigation and synthesis.

### PAct as a causal factor for health outcomes
The concept of PAct is often used to inform intervention development to support patient self-management and participation and engagement in healthcare.[19] The underlying assumption is that increases in PAct cause improvements in health outcomes. It is, therefore, important to understand not only whether there is an association between PAct and outcomes of diabetes and related metabolic disorders but also whether there is evidence for a causal pathway.

Two systematic reviews have assessed the impact of interventions targeting PAct on diabetes outcomes and found some evidence for effects on glycaemic control and self-management behaviours.[31 32] However, many of the included interventions are complex and include several components, and formal mediation analyses to assess whether interventions effects were mediated by increases in PAct were not carried out. It is, therefore, difficult to ascertain whether interventions effected change through PAct or other mechanisms.

Findings from individual studies suggest that PAct interventions can significantly decrease weight and blood pressure and improve glycaemic control in people with overweight or obesity[33] as well as reducing risk factors for CVD, such as smoking and lack of exercise.[34] However, to our knowledge, no systematic review has assessed the effects of PAct interventions for adults with overweight, obesity or CVD.

A systematic review of the literature is required to assess the association between PAct and outcomes of diabetes and related metabolic disorders and to critically appraise the strength of this evidence.

### AIMS
The aims of this review are:
1. To systematically review and synthesise evidence on the association between PAct and self-management behaviours relevant to diabetes and related metabolic disorders (eg, diet, physical activity).
2. To systematically review and synthesise evidence on the association between PAct and clinical outcomes of diabetes and related metabolic disorders (eg, blood pressure, HbA$_{1C}$).
3. To critically appraise whether the evidence is sufficient to assume a causal role of PAct in improving clinical outcomes and self-management behaviours.

### METHODS
The protocol is based on guidance on conducting systematic reviews provided by the Centre for Reviews and Dissemination,[35] Preferred Reporting Items for Systematic Reviews and Meta-Analyses (PRISMA)[36] and PRISMA Protocols.[37]

We will adopt a two-phase approach, whereby the first phase will involve a systematic scoping of the literature. This will involve establishing a list of all studies (cross-sectional, longitudinal, intervention) that examine the relationship between PAct and outcomes in our target population. Depending on the studies found in phase 1, we will then consider whether we are able to narrow down our review questions, for example, by population (eg, only diabetes populations) or study design.

### Inclusion/exclusion criteria
Studies will be eligible if they include a measure of PAct (eg, PAM, PACIC) and assess the association between PAct and clinical outcomes and/or self-management behaviours relevant to diabetes and related metabolic disorders, or if they assess the effect on such outcomes of interventions that explicitly target PAct.

### Population
We will include studies with samples consisting of adults (≥18 years old) who have diabetes or a related metabolic disorder. We defined 'diabetes and related metabolic disorders' to include pre-diabetes, diabetes (type 1/type 2 diabetes), obesity and CVD. We define pre-diabetes as a state with glycaemic levels above 'normal' but below cut-offs for a diagnosis of diabetes. As such, we will include any studies that describe their population as being

**Table 3** Categorisation of the suitability of different study designs (coupled with different analyses) to draw conclusions regarding a causal association between PAct and outcomes of diabetes and related metabolic disorders. PAct = Patient Activation

| Possible study designs+analyses | Suitability of study design and analyses | Rationale |
|---|---|---|
| RCTs with causal mediation analysis to assess whether PAct mediates intervention effects | Strong | RCTs are the only study design that allow causal mediation analysis to identify the mechanisms by which interventions exert their effects[51] |
| Cohort studies/RCTs or other intervention studies that assess the association between PAct and subsequent outcomes | Moderate | RCTs and longitudinal observational studies can provide temporal insights into the association between PAct and outcomes, which gives some indication of causality.[52] If an RCT examines the association between PAct and outcomes independent of study group allocation, randomisation has no bearing; analyses and findings are therefore akin to cohort studies. |
| RCTs that do not report on the association between PAct and outcomes but that show intervention effects on outcomes AND intervention effects on PAct, AND the intervention explicitly, mainly addresses PAct | Moderate | RCTs provide insight into causal effects of interventions on outcomes. If an intervention explicitly addresses PAct and there is evidence that the intervention influenced both PAct and outcomes, this provides indication for a causal mechanism of PAct on outcomes (though not definitive). |
| Observational cross-sectional studies | Weak | In cross-sectional designs, the time order of effects cannot be determined and therefore causality cannot be inferred.[53] |
| Intervention studies that are not RCTs (eg, pre-post studies) and that do not report on the association between PAct and outcomes but that show changes in outcomes AND changes in PAct. | Weak | Pre-post designs have the strength of temporality to indicate outcomes might be impacted by an intervention, but due to lack of randomisation causality cannot be inferred.[54] |

PAct, patient activation; RCT, randomised controlled trial.

diagnosed with pre-diabetes, impaired glucose tolerance, glucose intolerance, impaired fasting glycaemia, borderline diabetes, non-diabetic hyperglycaemia or similar.[38] We will not apply any specific criteria (eg, cut-offs for impaired fasting glucose or impaired glucose tolerance). We define CVD as any conditions affecting the heart or blood vessels, including (but not limited to): coronary heart disease (angina, heart attacks, heart failure), strokes and transient ischaemic attacks, peripheral arterial disease and aortic disease. Studies will be eligible if they include one or more of these disease types in a broader sample if results are reported separately for our population of interest.

### Interventions

We will include studies of varying designs, including intervention studies (see 'study designs'). Where we include intervention studies, any type of intervention will be eligible as long as PAct is measured and the study reports on its association with our predefined outcomes, since the primary aim of the review is not to assess the effectiveness of a particular type of intervention but to assess the relationship between PAct and outcomes.

If an intervention study reports intervention effects on PAct and effects on other specified outcomes but does not report on the association between PAct and outcomes, we will include the study only if (i) the intervention explicitly aims to increase PAct or is described as targeting patients' knowledge, confidence and skills for self-management (as opposed to interventions that target related but different constructs such as self-efficacy) and (2) increasing PAct is a key, main component of the intervention (ie, studies will be excluded if PAct components form part of a complex intervention with other components). Such studies will be excluded from quantitative synthesis but will be included in narrative synthesis as they can provide evidence of an association between PAct and outcomes.

### Comparators

Where we include intervention studies, any type of comparator will be eligible (as well as observational studies or other intervention studies with no comparator, for example, pre–post studies).

### Exposure

We will include only studies that include a measure of PAct (eg, PAM, PACIC or other measures of PAct). We will not include studies that measure related constructs (eg, confidence, or self-efficacy) if the measures do not explicitly purport to assess PAct.

### Outcomes

We will focus on clinical outcomes and self-management behaviours that are shared between diabetes and related metabolic disorders. Both self-reported and objectively

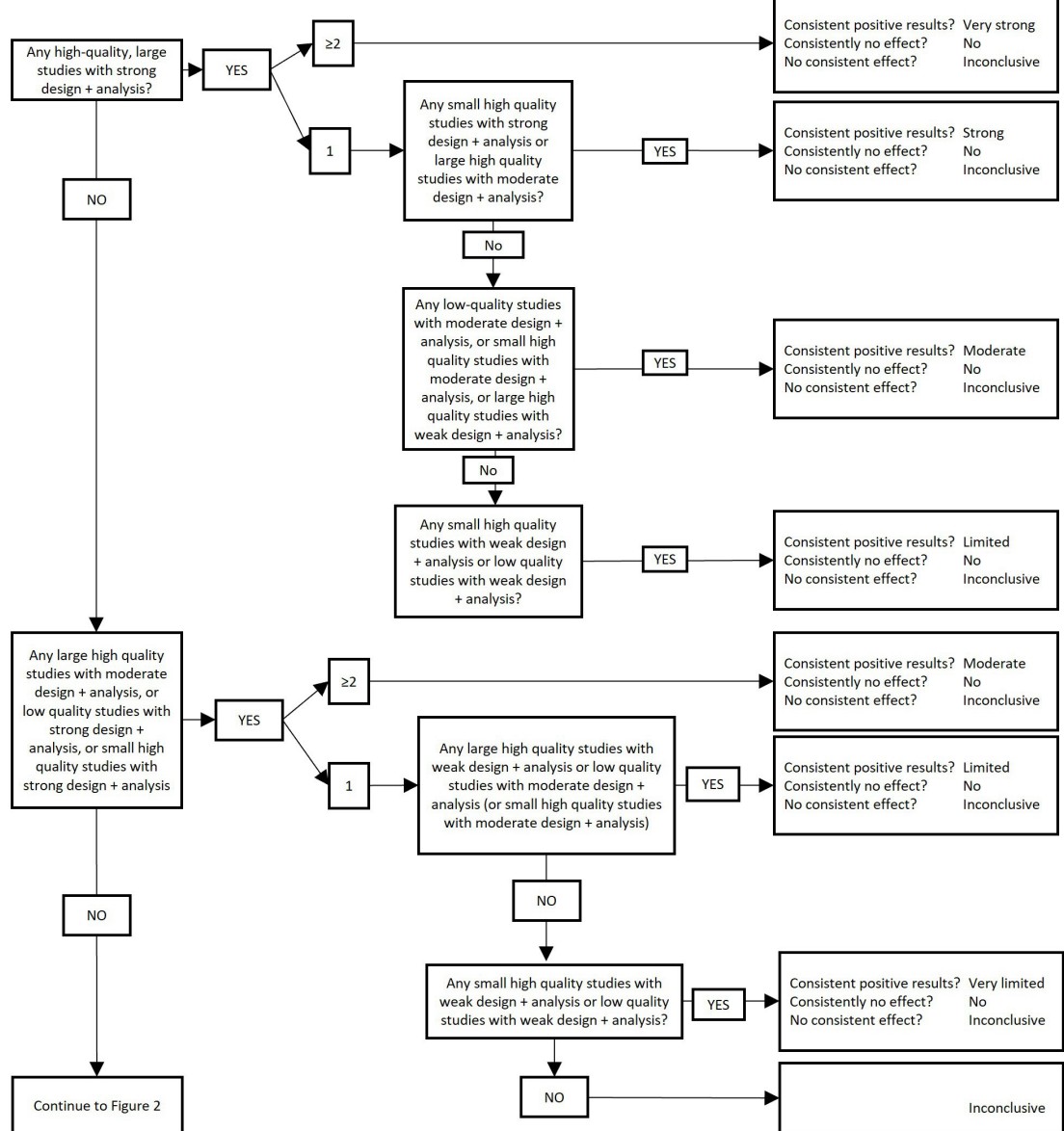

**Figure 1** Levels of evidence (part 1). To be used in conjunction with table 3 and figure 2. Note: studies including ≤250 participants or studies not providing sample size justifying a smaller sample size are considered 'small', studies including >250 participants are considered 'large'. Findings are considered consistent if at least two thirds (66.6%) of the highest quality studies are reported to have significant results in the same direction.

measured outcomes will be eligible. We will include studies that measure at least one of the following outcomes:

*Clinical outcomes*
► $HbA_{1C}$ level/glycaemic control.
► Systolic blood pressure/diastolic blood pressure.
► Low-density lipoprotein/ high-density lipoprotein/ total cholesterol.
► Serum triglycerides.
► Body mass index/body weight.

*Self-management behaviours*
► Outcomes related to diet (eg, fruit/vegetable consumption, following a low-fat diet)

► Outcomes related to physical activity (eg, step counts, following a regular exercise schedule, frequency of physical activity).
► Outcomes related to smoking (eg, smoking status).
► Outcomes related to alcohol consumption (eg, alcohol consumption, frequency or amounts).
► Medication adherence.

Study design

We will include original primary research articles. We will include all study designs, including cross-sectional, longitudinal and intervention (eg, randomised controlled trials (RCTs), pre–post comparison studies) as long as studies report on the association between PAct and one of the specified outcomes. We will exclude study protocols,

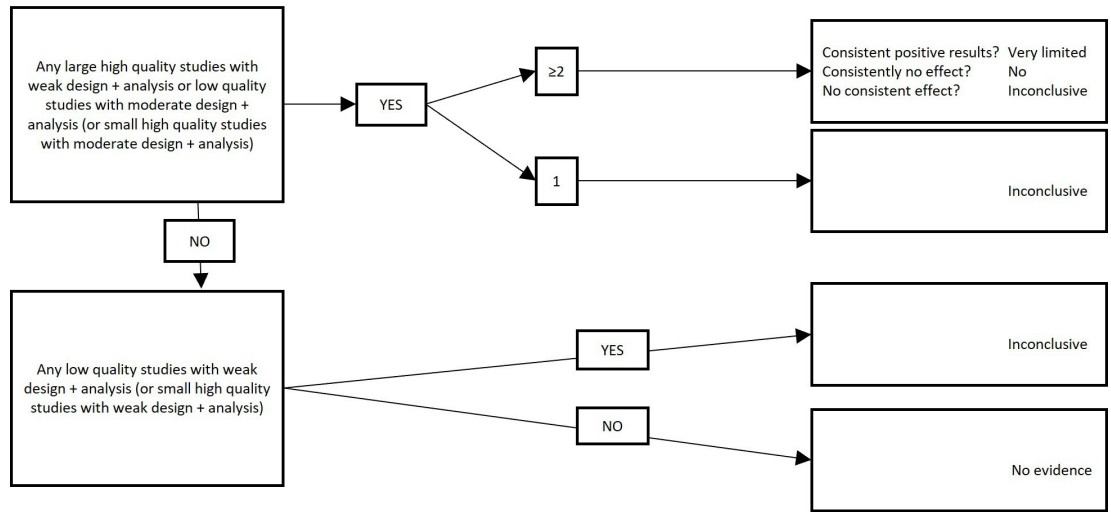

**Figure 2** Levels of evidence (part 2). To be used in conjunction with table 3 and figure 1. Note: studies including ≤250 participants or studies not providing sample size justifying a smaller sample size are considered 'small', studies including >250 participants are considered 'large'. Findings are considered consistent if at least two thirds (66.6%) of the highest quality studies are reported to have significant results in the same direction.

literature reviews/meta-analyses, qualitative studies and studies not reporting on empirical data.

### Language and date
We will include studies in any language, subject to local translation resources. Searches will not be limited by date.

### Publication status
We will endeavour to include both published and unpublished materials (eg, abstracts, theses) to reduce the impact of publication bias.[35]

### Information sources and search strategy
#### Databases
The following databases will be searched:
► Medline.
► Embase.
► CENTRAL.

► PsycInfo.
► Web of Science.
► CINAHL.

### Search strategy
The search strategy (table 1) was devised with the help of a medical librarian. The search strategy is outlined in table 1, and an example of the proposed search strategy is shown in online supplemental appendix A. References of included studies will be hand-searched for further eligible studies. Searches will be rerun prior to the final analysis. To identify relevant grey literature, we will search the Health Management Information Consortium (HMIC) database, ZETOC (using the conference search) and the British Library Integrated Catalogue.

**Table 4** Formulae to convert different measures of effect to Pearson's r, based on Wolf,[55] Friedman[56] and Hoeve *et al*[57]

| Statistic to be converted | Formula for transforming to pearson product moment correlation r | Notes |
|---|---|---|
| T | $\sqrt{\dfrac{t^2}{t^2+df}}$ | |
| F(df=1) | $\sqrt{\dfrac{F}{F+df_D}}$ | Use only for comparing two group means (df=1) $df_D$: df of the denominator |
| F(df >1) | $\sqrt{\dfrac{df_N(F-1)}{df_N+df_D}}$ | $df_N$: df of the numerator (k-1) $df_D$: df of the denominator (N-k) |
| $\chi^2$ (df=1) | $\sqrt{\dfrac{\chi^2}{n}}$ | Use only for 2×2 frequency tables (df=1) |
| $\chi^2$ (df >1) | $\sqrt{\dfrac{\chi^2}{\chi^2+N}}$ | |
| D | $\sqrt{\dfrac{d}{d^2+4}}$ | |
| Φ | (1) $\chi^2 = \phi^2 * N$ (2) Use equation for $\chi^2$(df=1) or $\chi^2$ (df >1) | |

**Table 5** Amendments to the protocol

| Date | Change | Rationale |
|---|---|---|
| 29 January 2021 | Removed 'Life expectancy/ total survival' from the list of outcomes | After discussion within the team, we decided this outcome does not align well with the other included outcomes. The other outcomes give an indication of how well people self-manage their condition, whereas life expectancy/survival is a wider measure that gives less insight into self-management specifically. Moreover, there are unlikely to be many studies with sufficiently long follow-up to provide any meaningful assessment of survival in this context, and even if there was a study with very long follow-up, we would then be relying on an assumption that the patient activation exposures measured at baseline do not change over time. |

## Data management and selection process

Citations returned through the database search will be exported into Covidence and deduplicated for screening. Two reviewers will independently screen titles and abstracts for eligibility and will then read full texts of selected citations to further assess eligibility. Any disagreements will be resolved by a third independent reviewer. Interrater reliability will be assessed using Cohen's Kappa.[39]

## Data extraction

Initially, we will extract study information into a table to summarise broad study characteristics. We will use this to assess the available evidence and decide whether to narrow down our review objectives (eg, to a specific disease population). Data from included studies will be extracted into a data extraction sheet (draft shown in online supplemental appendix B). The data extraction sheet is adapted from the Cochrane data collection form for RCTs and non-RCTs[40] and was also informed by the STROBE checklist of items that should be included in reports of observational studies,[41] the Consolidated Standards of Reporting Trials statement[42] and the risk of bias tools (RoB 2) we used (table 2).

Data to be extracted include details regarding study design, population, sample size, details about the intervention if relevant, methods used to assess outcomes, and details on the reported association between PAct and outcomes (including effect size, whether adjusted or unadjusted, and what covariates were included in adjusted models). One reviewer will extract data and one reviewer will independently check this for accuracy and completeness. The data extraction sheet will be pilot-tested by at least two reviewers on three studies. Any issues will be discussed and the sheet will be updated accordingly.

## Risk of bias/quality appraisal

We will use two different tools to assess ROB, depending on study design (table 2).

Each study will be appraised by two independent review authors. Reviewers will discuss any discrepancies until they reach a consensus, consulting a third reviewer if required. Any potential sources of bias or methodological limitations not covered by the tools will be noted by the reviewers. Each study will be assigned an overall risk rating of high, low or unclear (Risk of Bias Assessment Tool for Nonrandomised Studies; RoBANS tool) or high/ low/some concerns (ROB 2). ROB assessments will be used to determine the level of evidence (see the Levels of evidence). For the purpose of determining the level of evidence, ROB will be dichotomised into high/low risk (for RoBANS, 'unclear' and 'high' and for ROB2, 'some concerns' and 'high' will be amalgamated).

## Data synthesis and analysis

The study selection process will be depicted in a PRISMA diagram. Key results will be presented in form of a table summarising study characteristics. Risk of bias assessments will also be provided in a table.

### Narrative synthesis: levels of evidence

A key output of this review will be an assessment of the level of evidence available for assuming a causal association between PAct and self-management behaviours as well as clinical outcomes of diabetes and related metabolic disorders. The 'level of evidence' will be a composite measure, based on the strength of the study design/analysis, the quality of the study, sample size and the consistency of the findings, adapted from an approach used in a previous systematic review.[43]

Table 3 shows the types of study designs, coupled with different types of analyses, that could provide evidence for a causal assumption, grouped into different categories based on their suitability to support this assumption. If we encounter any unanticipated study designs/analyses, we will discuss this within the review team to assign the appropriate categorisation.

Once study designs and analyses have been categorised according to table 3 and once studies have been assigned a risk of bias appraisal, we will use figures 1 and 2 to assign a level of evidence, depending on the consistency of the findings across studies. Findings will be considered to be consistent if at least two-thirds (66.6%) of the highest quality studies are reported to have significant results in the same direction.[43]

### Narrative synthesis: harvest plot

If meta-analysis is not feasible and we cannot produce forest plots, we will create Harvest plots to synthesise and depict our findings, adapted from the approach used by Ogilvie *et al*.[44] The plot will consist of a matrix with one row per outcome, and one column (for the

assumption that there is a causal relationship between PAct and outcomes). Each study will be represented by a bar in each row for which that study reported relevant evidence. The strength of the study design and the analysis will be represented by the height of the bar, with higher bars indicating more suitable design and analysis. Studies using self-reported outcomes will be represented by a grey bar, while bars for studies using objective measures will be black. Each bar will be annotated with the quality appraisal for that study (eg, high, low or unclear) and the sample size.

## Meta-analysis

Meta-analysis will be undertaken if studies are considered sufficiently similar in their research questions, designs and outcomes. From each study, we will extract effect sizes for the association between PAct and the prespecified outcomes. We will extract unadjusted and adjusted associations, and synthesise these separately. Regression coefficients from models with different sets of covariates represent different parameters and cannot be combined meaningfully.[45] We will, therefore, initially assess which covariates are included in adjusted models and, if there is agreement between models in terms of key covariates, we will synthesise coefficients across models (even if model specifications are not completely identical). If there is insufficient agreement between models in terms of covariates, we will include adjusted associations in the narrative synthesis and focus on unadjusted associations in the quantitative synthesis.

We expect studies to report a wide range of different estimates of the association between PAct and outcomes. We will, therefore, initially convert different measures of the association to the Pearson Product Moment Correlation using the formulae in table 4, because the correlation coefficient is an easily interpretable effect size to assess the strength of association between two variables. Some studies may report only ORs (as PAct scores are often dichotomised into high/low and clinical outcomes are often dichotomised into within/not within normal range). If studies report ORs, we will construct contingency tables based on information about percentages of PAct levels and outcomes and use these tables to calculate $\chi^2$ values, which can then be transformed to r.

We will use a random-effects approach, because we assume that the population effect sizes vary randomly from study to study (rather than assuming the population effect size is the same for all studies), eg, due to differences in age, socioeconomic status, geographic location or disease. Random effects meta-analysis allows inferences beyond the studies included in the analysis.[46] However, if the number of included studies are ≤5, we will also perform a sensitivity analysis with a fixed-effect approach. This is because when heterogeneity is present, a random-effects meta-analysis weights the studies relatively more equally than a fixed-effect analysis, and, thus, small-study effects could bias the findings.

We will use the Hunter-Schmidt random-effects method to synthesise correlations across studies, because this method produces more accurate estimates than the Hedges-Olkin and Rosenthal-Rubin methods (except when the average population effect size is very large).[46] Effect sizes from cross-sectional and longitudinal studies will be synthesised separately.

If a study reports more than one estimate of association for a particular combination of exposure and outcome, we will select the estimated association based on the longest duration of follow-up or the most precise measure of the outcome. If it is not possible to discern this, within-study meta-analytic calculations will be used to obtain a single effect size, to maintain the statistical assumption of independence necessary for a meta-analysis. If the effect sizes are based on different sample sizes, the average sample size will be calculated and used for subsequent analyses.

### Exploration of heterogeneity

If sufficient studies are available, we will perform meta-regression to assess whether the effect size varies with study characteristics, including:

▶ Studies with different populations (diabetes/pre-diabetes, obesity, CVD).
▶ Self-reported versus objectively measured outcomes.
▶ Clinical versus behavioural outcomes.

Meta-regression will be performed on correlations transformed according to the Fisher z-transformation.[47]

### Sensitivity analyses

Sensitivity analysis will be performed excluding studies that are categorised as high risk of bias, to assess whether findings are unduly influenced by these studies.

## Assessment of heterogeneity and reporting bias

To assess heterogeneity, we will report the $I^2$ statistic with a 95% confidence interval as well as outcomes from the test for heterogeneity (Q-statistic and associated p value). For $I^2$, we will categorise heterogeneity as low (0%–30%), moderate (30%–60%), substantial (60%–90%) and considerable (90%–100%).[48] To assess publication bias, we will construct funnel plots, plotting the mean correlation against study sample sizes as well as the residual SD of r against the sample size.

## PATIENT AND PUBLIC INVOLVEMENT

We shared a lay summary of the review protocol with an established patient and public involvement (PPI) panel. Feedback was positive, with panel members commenting that they think the review will be useful, particularly within NHS services. Panel members also made recommendations for our dissemination strategy to help us reach a wider audience. After completing the review, we will seek feedback from the PPI panel on a lay summary of the review findings and on our dissemination plan. The protocol was further reviewed by a General Practitioner (GP) partner from NHS Cambridgeshire and

Peterborough Clinical Commissioning Group (CCG), who has particular expertise in person centred, collaborative care and long-term conditions.

## ETHICS AND DISSEMINATION

Ethical approval is not required for this systematic review. The review will be disseminated in the form of a peer-reviewed journal article, at conferences and other presentations (eg, webinars) as well as more publicly accessible formats such as blog posts, social media posts and, if suitable, a press release. The findings of the review will be of interest to clinical commissioning groups, policymakers and intervention deliverers/developers that currently use, or plan to use, the PAM or other measures of PAct to tailor and allocate interventions for diabetes and related metabolic disorders. It will also be of relevance to those using measures of PAct to evaluate intervention effectiveness and healthcare performance, as it will provide an indication of how well PAct predicts outcomes for diabetes and related metabolic disorders.

## AMENDMENTS

Amendments made will be noted in a prespecified section of the protocol (rather than being incorporated into the protocol), with the date and rationale. Amendments will also be uploaded to Prospero. Since commencing title/abstract screening, we have made one amendment (table 5).

**Acknowledgements** We would like to thank Dr Isla Kuhn for reviewing our protocol and helping us refine our search strategy. We also thank our PPI panel and Dr Mark Brookes for reviewing our protocol and providing feedback and comments.

**Contributors** JM drafted the manuscript, with regular input from all co-authors. All authors read, provided feedback and approved the manuscript prior to submission. JM, AA, SJG, RR, JMB and AD contributed to the development of the selection criteria, the risk of bias assessment strategy and data extraction criteria. SS provided statistical expertise.

**Funding** This work was supported by the Medical Research Council [grant number MC_UU_00006/6] and the National Institute for Health Research (NIHR) under its Programme Grants for Applied Research Programme (RP-PG-0216–20010).

**Competing interests** JM and RR are Trustees for the Association of the Study of Obesity (unpaid roles). AA and SJG are the chief investigators on two publicly funded (MRC, NIHR) trials where the intervention is provided by WW (formerly Weight Watchers) at no cost outside the submitted work. All other authors report no competing interests.

**Patient consent for publication** Not applicable.

**Provenance and peer review** Not commissioned; externally peer reviewed.

**ORCID iDs**
Julia Mueller http://orcid.org/0000-0002-4939-7112
Amy L Ahern http://orcid.org/0000-0001-5069-4758
Stephen J Sharp http://orcid.org/0000-0003-2375-1440
Rebecca Richards http://orcid.org/0000-0001-7122-6822
Jack M Birch http://orcid.org/0000-0001-6292-1647
Alan Davies http://orcid.org/0000-0001-5737-5629

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
