## [Reviewer comments · BMJ Open]

ARTICLE DETAILS

TITLE (PROVISIONAL)	The association between patient activation, self-management behaviours and clinical outcomes in adults with diabetes or related metabolic disorders: A systematic review and meta-analysis protocol
AUTHORS	Mueller, Julia; Ahern, Amy; Sharp, Stephen; Richards, Rebecca; Birch, Jack; Davies, Alan; Griffin, Simon

VERSION 1 – REVIEW

REVIEWER	Ji, MEihua Capital Medical University School of Nursing
REVIEW RETURNED	05-Oct-2021

GENERAL COMMENTS	The current study reported the protocol in examining the association of patient activation, self-management behaviors and clinical outcomes among adults with metabolic disorders through a systematic review and meta-analysis. The protocol addressed a significant issue in these patients' population and searched 6 databases for related evidences, along with other resources. Although authors pointed out the current gap and necessities in conducting the systematic review and meta-analysis, there is lack a of clarity on the definition of patient activation (for example, is there a theoretical framework that explains the extent of the construct?). In addition, one of the aims is to examine whether a causal link between PAct and outcomes existed, however, the protocol will include cross-sectional studies, this will not infer causal relationships between study variables; therefore suggest removing studies with cross-sectional design.
---

REVIEWER	Jang, Yeonsoo Yonsei University
REVIEW RETURNED	14-Oct-2021

GENERAL COMMENTS	This manuscript seems to be significant. The research process is clear. However, I think populations need to be narrow down. There are the differences of the goals in self-management behaviors among DM and CVD. Therefore, I recommend that the populations include preDM, DM, and metabolic disorder except CVD.
--

REVIEWER	Dintsios, Charalabos-Markos Heinrich Heine University Düsseldorf
REVIEW RETURNED	31-Oct-2021

GENERAL COMMENTS	A very well structured and written protocol for a systematic review including potential meta-analysis. I have only a few comments, which probably will ameliorate the manuscript. Abstract: With regard to the assessment of risk of bias I would name the instruments already in the abstract if possible. Strengths and limitations of this study: P3L4: If it is meant that the performed search strategy is to be hypersensitive pls. state this directly. Background: P3L27: Some of the referred literature on the direct and indirect costs are rather old. There is some more actual literature available for Germany as well: Bächle C, Claessen H, Andrich S, Brüne M, Dintsios CM, Slomiany U, Roggenbuck U, Jöckel KH, Moebus S, Icks A. Direct costs in impaired glucose regulation: results from the population-based Heinz Nixdorf Recall study. BMJ Open Diabetes Research and Care, 2016; 4:e000172.doi:10.1136/bmjdr-2015-000172 Population: P6L33: I am not sure that the inclusion of gestational diabetes suits well, since this is a very special condition correlated with potential later onset of diabetes mellitus and patient activation of pregnant women in the sense of primary prevention is probably not captured well with the interventions under investigation. Databases: P8L50: CRD-HTA database might be also relevant Meta-Analysis: P15L7et seq.: I would propose to the authors depending on the identified numbers of studies to perform additionally a fixed effect model as a sensitivity analysis, especially in cases where included studies are less than or equal 5. P15L31: reference not found. Figure 1: Could be simplified by defining in general inconsistent effect(s) as inconclusive and thereby avoiding to repeat this condition all the time in the flow chart. The same holds for consistently no effect. Furthermore, one cited reference is not found.
---

VERSION 1 – AUTHOR RESPONSE

Reviewer Reports:

Reviewer: 1

Dr. Meihua Ji, Capital Medical University School of Nursing

Comments to the Author:

The current study reported the protocol in examining the association of patient activation, self-management behaviors and clinical outcomes among adults with metabolic disorders through a systematic review and meta-analysis. The protocol addressed a significant issue in these patients'

population and searched 6 databases for related evidences, along with other resources.

RESPONSE:

Thank you for your helpful comments and for your time and consideration. We appreciate it.

Although authors pointed out the current gap and necessities in conducting the systematic review and meta-analysis, there is lack a of clarity on the definition of patient activation (for example, is there a theoretical framework that explains the extent of the construct?).

RESPONSE:

We have added some clarification to better define and describe patient activation:

“The construct encompassing patients’ knowledge, confidence and skills for self-management has been termed ‘patient activation’ (PAct).[16] Consumer driven health care approaches and many chronic illness care models assume that more “activated” patients (i.e. patients with the relevant knowledge, confidence and skills to self-manage their own health and healthcare) will play a more active role in managing their health and have better health outcomes [16]. Conversely, less “activated” patients are expected to be less likely to see out help, adhere to medical advice, and manage their own health.” (p. 3, lines 20-26)

In addition, one of the aims is to examine whether a causal link between PAct and outcomes existed, however, the protocol will include cross-sectional studies, this will not infer causal relationships between study variables; therefore suggest removing studies with cross-sectional design.

RESPONSE:

The aim is specifically “To critically appraise whether the evidence is sufficient to assume a causal role of PAct in improving clinical outcomes and self-management behaviours.”

Therefore, we will include all study designs and then critically appraise whether the evidence base is sufficient to assume a causal role of PAct. If, for example, we find that the current evidence base includes mostly cross-sectional studies, we will conclude that there is insufficient evidence to assume a causal relationship. This is also highlighted in Figure 1 and Table 3, where we show how we will appraise studies and how we will decide if the evidence base is sufficient. In Table 3, for example, we have clearly highlighted that evidence from cross-sectional studies will be considered “weak” because “In cross-sectional designs, the time order of effects cannot be determined and therefore causality cannot be inferred”. There is therefore no need to exclude cross-sectional studies – we will include all evidence and then assess the strengths/weaknesses.

Reviewer: 2

Dr. Yeonsoo Jang, Yonsei University

Comments to the Author:

This manuscript seems to be significant. The research process is clear.

RESPONSE:

Thank you for your helpful comments and for your time and consideration. We appreciate it.

However, I think populations need to be narrow down. There are the differences of the goals in self-management behaviors among DM and CVD. Therefore, I recommend that the populations include preDM, DM, and metabolic disorder except CVD.

RESPONSE:

Thank you. We agree that the population is very broad. This is because the availability of evidence was unclear prior to us commencing our review, and we therefore kept the population broad to begin

with. We included CVD because hyperglycaemia is a risk factor for CVD, and there is considerable overlap in the self-management strategies for each of these stages along the disease trajectory from normoglycaemia to the cardiovascular complications of hyperglycaemia eg relating to diet, smoking cessation, alcohol consumption, physical activity and medication adherence.

As stated in the Methods section, "We will adopt a 2-phase approach, whereby the first phase will involve a systematic scoping of the literature. This will involve establishing a list of all studies (cross-sectional, longitudinal, intervention) that examine the relationship between PAct and outcomes in our target population. Depending on the studies found in Phase 1, we will then consider whether we are able to narrow down our review questions, e.g. by population (e.g. only diabetes populations), or study design."

Since submitting our protocol, we have completed the scoping of the literature. Following an assessment of the available evidence, we have determined that we will be able to narrow our population down to those with Type 2 diabetes only. Therefore, our review will be more focused and will have a more homogenous population. However as the protocol should reflect our actual process, we prefer to keep the protocol as is. We will then clarify when we write up our review that we took this scoping approach and then narrowed down our population.

Reviewer: 3

Dr. Charalabos-Markos Dintsios, Heinrich Heine University Düsseldorf

Comments to the Author:

A very well structured and written protocol for a systematic review including potential meta-analysis. I have only a few comments, which probably will ameliorate the manuscript.

RESPONSE:

Thank you for your helpful comments and for your time and consideration. We appreciate it.

Abstract: With regard to the assessment of risk of bias I would name the instruments already in the abstract if possible.

RESPONSE:

Thank you for this suggestion. We have added this information to the abstract:

"Two reviewers will independently screen titles/abstracts and full texts and assess risk of bias using the revised Cochrane risk-of-bias tool for randomized trials (RoB 2) or the Risk of Bias Assessment Tool for Nonrandomized Studies (RoBANS)." (p. 2, lines 17-18)

Strengths and limitations of this study:

P3L4: If it is meant that the performed search strategy is to be hypersensitive pls. state this directly.

RESPONSE:

We do not think the term "hypersensitive" applies here (nor 'Cochrane Highly Sensitive Search Strategy' as this only applies to reviews on randomised trials as far as we are aware). The line referenced is "It [the review] uses a comprehensive search strategy with a broad range of relevant databases". We think this is an accurate description.

Background:

P3L27: Some of the referred literature on the direct and indirect costs are rather old. There is some more actual literature available for Germany as well: Bächle C, Claessen H, Andrich S, Brüne M, Dintsios CM, Slomiany U, Roggenbuck U, Jöckel KH, Moebus S, Icks A. Direct costs in impaired glucose regulation: results from the population-based Heinz Nixdorf Recall study. *BMJ Open Diabetes Research and Care*, 2016; 4:e000172.doi:10.1136/bmjdr-2015-000172

RESPONSE

We have added the suggested reference. We have also removed the two older papers and added a further newer reference (Ryder S, Fox K, Rane P, et al. A Systematic Review of Direct Cardiovascular Event Costs: An International Perspective. *Pharmacoeconomics* 2019 377 2019;37:895–919. doi:10.1007/S40273-019-00795-4). (p. 3, line 7)

Population:

P6L33: I am not sure that the inclusion of gestational diabetes suits well, since this is a very special condition correlated with potential later onset of diabetes mellitus and patient activation of pregnant women in the sense of primary prevention is probably not captured well with the interventions under investigation.

RESPONSE:

We have removed gestational diabetes from our population criteria. (p. 6, line 28)

Databases:

P8L50: CRD-HTA database might be also relevant

RESPONSE:

Thank you for this suggestion. We have had a look at the HTA database and have run some searches using our search terms. However this returned only two studies, both of which were not eligible for the review. We have therefore decided not to add this to our list of databases.

Meta-Analysis:

P15L7et seq.: I would propose to the authors depending on the identified numbers of studies to perform additionally a fixed effect model as a sensitivity analysis, especially in cases where included studies are less than or equal 5.

RESPONSE:

Thank you for the suggestion. We have edited the Meta-analysis section:

“However, if the number of included studies is ≤ 5 , we will also perform a sensitivity analysis with a fixed-effect approach. This is because when heterogeneity is present, a random-effects meta-analysis weights the studies relatively more equally than a fixed-effect analysis, and thus small-study effects could bias the findings.” (p. 15, lines 8-11)

P15L31: reference not found.

RESPONSE:

Apologies, we believe we have now fixed this issue.

Figure 1: Could be simplified by defining in general inconsistent effect(s) as inconclusive and thereby avoiding to repeat this condition all the time in the flow chart. The same holds for consistently no effect.

RESPONSE:

We tried moving this as suggested but did not find that this improved the readability of the figure. In one option we removed the no effect/inconsistent effects from the boxes and added it as one separate box as an additional step in the flowchart. The Figure was then shorter (vertically) but broader horizontally. We also tried moving this information into the caption but felt this would be harder for the reader to follow. We have therefore opted to keep the figure as is.

Furthermore, one cited reference is not found.

RESPONSE: Apologies, this error was caused by Word's cross-referencing function; we have now removed this for all tables so the issue should be resolved.

VERSION 2 – REVIEW

REVIEWER	Dintsios, Charalabos-Markos Heinrich Heine University Düsseldorf
REVIEW RETURNED	04-Jan-2022

GENERAL COMMENTS	The authors have implemented almost all my proposals and the manuscript has been ameliorated significantly.
---